# Soft Passing over Traffic-Calming Devices by Controlled Suspension in Low-Speed Robotic Vehicles for Vulnerable People

**Ricardo Zavala-Yoé [1,*,†], Miguel Sandoval-Olivares [1,†], Luis Carlos Félix-Herrán [2,*,†] and Ricardo A. Ramírez-Mendoza [3,†]**

1. Tecnológico de Monterrey, Escuela de Ingeniería y Ciencias, Calzada del Puente 222, Col. Ejidos de Huipulco, Mexico City 14380, Mexico; saol.miguel@gmail.com
2. Tecnológico de Monterrey, Campus Sonora Norte, Blvd. Enrique Mazón López 965, Hermosillo 83000, Mexico
3. Tecnológico de Monterrey, Av. Eugenio Garza Sada 2501 Sur, Tecnológico, Monterrey 64849, Mexico; ricardo.ramirez@tec.mx
*   Correspondence: rzavalay@tec.mx (R.Z.-Y.); lcfelix@tec.mx (L.C.F.-H.)
†   These authors contributed equally to this work.

**Abstract:** The usefulness of golf carts for transporting patients in hospital facilities is well known. Nursing homes, medical campuses, and any type of related service require the low-speed transport of patients either in a seat, in a wheelchair, or on a stretcher. This type of transport is not limited to hospitals, but also includes other environments where there are people with special requirements. Think for instance of handicapped or elderly people that need a van because they have to go from their homes to any destination; therefore, the use of golf carts becomes relevant and attractive. Moreover, these carts could be automated for path following and deal with bumps, potholes, or sinkholes. In this context, the present research proposes a novel way to deal with this kind of road obstacle when the gentle transport of patients is a key element. In order to pass over these obstacles, a soft upwards displacement of the front and rear sections of the vehicle was achieved with magnetorheological dampers as part of the vehicle's suspension system. In this way, people who need this gentle transport will not have any discomfort. Moreover, this work is aligned with the spirit of Automated Vehicles 3.0.

**Keywords:** automated car/golf car suspension; non-linear control; special-needs people transport

## 1. Introduction

Currently, there are various devices that comprise the suspension system of a vehicle. Active and semi-active systems use control techniques for two objectives, passenger comfort and vehicle stability. In the literature, it can be found that from varied modeling and control perspectives [1] to classical control policies, these viewpoints focus on only one of these two objectives. Similarly, there are techniques that seek to ensure both of them. In [2], the control techniques Skyhook, Groundhook, and Mix-1-sensor for a semi-active suspension were compared. Such techniques are on–off, this is they only have two values corresponding to a soft or rigid damping force. Skyhook uses an imaginary damper between the body and the sky, trying to minimize the vertical acceleration of the vehicle body, but the contact between the tires and the ground does not provide an acceptable stability. Although this control strategy was proposed in 1974 by D. Karnopp [3], it is a reference in control laws focused on comfort. Groundhook is similar to Skyhook, but focuses on the vehicle stability, reducing the dynamic forces between the tires and the surface using a virtual damper that connects the tire and the road. Mix-1-sensor minimizes the acceleration of the sprung mass by measuring the oscillation frequency of the suspension and thereby determines a high or low damping.

## 2. Definitions and Vehicle Categories

Originally, low-speed vehicles emerged as golf carts for transportation purposes, but today, a variety of applications have appeared as a result of their versatility and ease of use. Thus, these carts are also requested for maintenance, landscaping, and even security and medical services as short-range transportation within hospital campuses. Notice that although golf carts may seem harmless, it is important to realize that this type of transportation can be very dangerous when used improperly and without caution or expert supervision.

It is important to remark that a golf cart is understood as a motor vehicle that is intended for use primarily on golf courses or on roadways where the access and use of other motor vehicles is controlled. Moreover, these cars [4]:

- Have an electric motor or internal combustion engine that is:
- Incapable of propelling the vehicle at a rate of speed of 32 km per hour or greater on a level surface or;
- Speed-modified to prevent the vehicle from attaining a rate of speed of 32 km/h or greater on a level surface.

A golf car or golf cart (GC) (Strictly speaking, cars are self-impelled, but carts are not. Some times, there is an abuse in denomination, and both terms are considered as synonyms, although they should not [5].) is a generic denomination for many types of vehicles. Technically, golf carts are vehicles that are specifically used on golf courses and reach top speeds of less than 24 km/h (15 mph). However, if a vehicle can develop speeds between 24 km/h and 32 km/h (15 mph and 20 mph, respectively), the car is considered as a personal transportation vehicle (PTV). There is a third type of car, and it is referred to as a low-speed vehicle (LSV). An LSV is a street-legal, four-wheeled electric vehicle with a top speed of 25 mph (40 km/h) and a gross vehicle weight rating of less than 3000 lb (1360 kg). In the US, most states allow LSVs to drive on roads marked 35 mph or less, but there may exist variations [6,7]. They are designed for local trips in areas such as planned communities, resorts, school campuses, etc. In the U.S., these motor vehicles are regulated by the National Highway Traffic Safety Administration (NHTSA) [7,8].

These transport vehicles can be operated with gasoline or electricity, and although they can be driven on public roads, GCs and PTVs are not classified as motor vehicles according to federal law (in the U.S. and Canada). In spite of variations in the law, a GC is intended for golf courses in agreement with ANSI Z130.1; PTVs are intended for designated roadways in line with ANSI Z135, and LSV are for public roads (FMVSS 500) [8].

In the U.S., most states allow GCs and LSVs to operate on roadways with a maximum speed limit of 56 km/h or 35 mph. However, in contrast, Texas and Alaska allow in certain places up to 71 km/h (45mph). In Table 1, a short comparison between GCs and LSVs is shown. PTVs and other types of these vehicles were excluded for simplicity. As a result of variations in the law and depending on the U.S. state regulation, safety features such as seat belts, mirrors, tail lamps, etc., are subject only to state or local safety equipment requirements [7]. Although golf carts are allowed on streets and highways with posted speed limits, this study focused on transportation for medical purposes or vehicles for the elderly within the circuits of a hospital campus. In spite of some differences found in the literature, Table 1 is offered here to contrast the main issues between GCs and LSVs [4,6–9]. In addition, most states require a valid driver's license and insurance for all GCs and LSVs on public roads. For instance, Florida does not require a golf cart operator to hold a driver's license, but they may require a minimum age (typically 16) for legal operation of a GC on public roads.

European regulations on GC vehicles vary as well. For instance, in [10], it was specified that all GC vehicles should now be approved at the European level. An electric GC on public roads has a power of 2.2 kW (23–30 km/h), and an LSV has a power of 5.5 kW (40–45 km/h). As of 1 January 2017, all new golf carts L7e registered in Europe should

comply with the new European regulation (2017). Furthermore, in [11], an additional class of these vehicles was also investigated in terms of driving regulations.

**Table 1.** Main differences between GCs and LSCs.

| Feature | LSV | GC |
|---|---|---|
| Passengers | 2–6 | 2–4 |
| Comfort char. | Smooth ride | - |
| Regulations | Street legal (var.in law) | Not allowed on streets |
| Safety aspects | Seat belts | - |
| Speed | 40 km/h (25 mph) | $v < 24$ km/h (15 mph) |
| Weight | Up to 590 kg (1300 lb) | Up to 1350 Kg (3000 lb) |
| Location | Public roads | Golf course |
| Safety standard | ANSI Z130.1 | FMVSS #500 |

## 3. Low-Speed Specialized Transport for Disabled and Elderly People

As mentioned above, in the U.S., the NHTSA [12] considers that the elderly is a group in need of special transport systems. The first step to assist older people is by means of what is known as paratransit or shared transport services. These alternative vehicles help the elderly in situations that prevent them from using public transportation. Thus, most of these services provide door-to-door attention from the passenger's home to their destination, eliminating the need to board a bus. Moreover, this transport necessity goes beyond people belonging to this group who need or choose more personal attention [9,11].

For instance in the U.S., the University of South Florida Institute on Aging funded a project to establish a baseline of information about the use of "neighborhood electric vehicles" (NEVs) by older people and others who do not have access to personal vehicles or that, for any reason, select to use GCs [12]. These GC-NEVs (or NEVs) are sold by enterprises such as Bombardier NV, Club Car, Columbia Par Car, E-Z-GO, Hyundai, and Yamaha, among others. Besides, NEVs carry from 2–4 adults. Something very relevant here is that in some U.S. states (for instance, Florida), a golf cart operator is excused from holding a driver's license; so, the operator is allowed to operate a GC along roadways designated for these carts [9,12]. The latter has been the subject of intense conflicts between local and federal laws in the U.S. because there also exist modified CG-type vehicles that people wish to use on/off roads. An interesting additional classification-based regulation was presented in [13] that solves the above-mentioned problems.

As a consequence of these transport issues and the successful adaptation of basic GCs, LSVs, and PTVs to a wide range of needs, it is possible to focus on the elderly, as well as on other vulnerable classes that need to be considered. More specifically, this research refers to the group of people who suffer from a physical or mental disability, have aged, or both.

In Europe, the use of GCs is also very widespread. Transportation in hospital circuits is very much required, and patients or visitors can request their service (see for instance a couple of cases in The Netherlands [14,15]). For example, in Germany, the Red Cross represents an additional resource to help elderly and handicapped people with such transportation [16,17]. A complete report about many types of vehicles and physical characteristics was published in Austria and considers many vehicles [18]. Wider perspectives within Europe and other parts of the world can be consulted in [19]. Despite the existence of regulations (local or federal) in developed countries, GCs are often improvised for use in particular settings. However, even these particular environments are well paved and well finished. This fact is not necessarily the rule in the case of streets, paths, and even more in hospital and university circuits in developing countries or Eastern Europe.

### 3.1. Golf Cars in Developing Countries

As mentioned above, a car can be considered a GC if the average speed of the vehicle is less than 24 km/h and constrained to move on a horizontally level surface composed

of a concrete surface or dry asphalt and free of loose material or surface contamination with a minimum coefficient of friction of 0.8 between the tire and the surface [5]. The latter was established by the International Light Transportation Vehicle Association (ILTVA) in the U.S. [5]. The ILTVA is the Standards Developer Organization (SDO), and it is an accredited standards developer organization under ANSI (The American National Standards Institute (ANSI) is the U.S. administrator for voluntary standardization. At the same time, it is the official U.S. representative of the International Standards Organization (ISO) [5].) and is sponsored by Yamaha, E-Z-GO, and Club Car, which are the most important golf car assemblers in the U.S. and the world. The ILTVA also has a representation in the Americas [5]. In spite of these organizations and regulations, the operation of a golf cart may not require the possession of a driver's license as a generalized standard (at least in the U.S.), making them a viable option for persons who do not drive automobiles [7,20]. For instance, in Germany, it is required from the GC operator to hold a valid driver's license if the driver has to travel on public streets, and the same situation happens in Ontario, Canada [4]; however, if the vehicle is intended for confined environments such as farms, industries, and forests, such a permit is not required [21].

If regulations in developed countries are a complex process, it is even more complicated in developing nations. For instance, although in [22], the prohibition of transporting people in those carts was clearly established, this is the only mention in this regard. As in the U.S., local (or even, particular) laws seem to govern this matter. It is remarkable that golf clubs provide specific rules for the use of GCs [23–25]. In a certain region of Colombia, it is recognized that GCs are not even defined in the National Code of Transit, but it is established that every motorized vehicle must have a national registration. However, in [26], it was explained that if, according to the manufacturer, GCs are not intended for on-street use, then official registration is not necessary.

*3.2. Traffic-Control Devices*

As a result of the information described above about the variation in regulations and driver's license permit requirement, it is natural to expect that additional safety measures have to be taken into account. In this spirit, unsurprisingly, driving a GC off-road can trigger the desire to play with speed. Moreover, either by imprudence or overconfidence, accidents may occur.

It is reported that these types of vehicles are involved in accidents each year. Such events may result in personal injury, damage to federal, state, or neighborhood property, or even death. GCs and PTVs are not as good as LSVs in aspects such as maneuverability, stability, and safety features [5,20,27]. It is important to list some important safety issues that have to be considered when driving GCs and PTVs (traveling downhill was ignored in this study):

- Tip over. Lightweight with respect to the transported weight of the passengers (see Table 1), a high center of gravity, and small tires result in that these cars may be highly unstable. This risk increases if they are driven on rough and uneven terrain;
- Safety accessories. Not all regulations require that these cars come equipped with seat belts or restraints;
- Absence of side doors. By design, the lack of side doors can cause accidents.

Trying to prevent or diminish these risks, what are known as traffic-control devices have been developed. As established by the (U.S.) Manual on Uniform Traffic Control Devices (MUTCD) (The Manual on Uniform Traffic Control Devices for Streets and Highways defines the standards used by road managers (U.S.) nationwide to install and maintain traffic-control devices on all public streets, highways, bikeways, and private roads open to public travel. The MUTCD is published by the Federal Highway Administration (FHWA) [28].), the purpose of traffic-control devices, as well as the principles for their use are to promote highway safety and efficiency by providing for the orderly movement of all road users on streets, highways, bikeways, and private roads open to public travel throughout the nation.

As a result of many complaints against drivers in The Netherlands, the idea of a device referred to as a *traffic calming* was developed. The first device of this nature was built and installed in Delft in 1970 when city officials built a 0.26 ft (8 cm) road hump at the end of an alley [29]. Since these measures were successful, the use of such devices extended through—practically—all of Europe [30].

Regulations in the U.S. define a traffic-calming device as the combination of mainly physical measures that reduce the negative effects of motor vehicle use, alter driver behavior, and improve conditions for non-motorized street users [31]. Traffic-calming devices are then used to decrease speed or the volumes of cars along residential streets and may be classified as either volume- or speed-control devices [30].

As described in the Abstract, the goal of this manuscript was to propose and simulate a semi-active suspension that raises a commercial vehicle chassis so as not to damage the underside when the car passes through a speed-calming device. In places such as Mexico City, it is common to find bumps, holes, pot holes, roads in poor condition, etc. They can damage the vehicle suspension, and in many cases, they can lead to catastrophic accidents. Lifting the car structure 10–12 cm above the top of the bump will help with this. The non-linearity of the suspension structure, unknown car load, and exact knowledge of the path and speed-calming device make the problem of finding an adequate control a complicated task.

### 3.2.1. The Case of Speed Bumps and Speed Humps

Although giving a complete exposition of the types and regulations of traffic-control devices is out of the scope of this manuscript, it is important to emphasize the class of artifacts that is considered in the modeling and control action for the suspension system herein. It is planned that the car's frame elevates to avoid a drastic pass over speed-control devices, in particular speed bumps and speed humps. These artifacts, actually speed-control measures, are defined as physical devices planned to decrease vehicle speed. Some of these devices have also been shown to have an impact on traffic volumes [30]. Thus, speed-control devices can be divided into:

- Horizontal measures. These are physical equipment that require vehicles to shift laterally. This fact obliges drivers to minimize speed to comfortably maneuver through and around the shift;
- Narrowing. This happens when the travel lane is physically reduced;
- Vertical measures. These are artifacts created to vertically displace the chassis of a vehicle. In this way, the driver is forced to reduce speed to comfortably cross this type of obstacle. Examples of them are speed tables, speed bumps, and speed humps.

It has been remarked that speed humps have progressed from detailed and ongoing investigation to practical and statutory implementation in Europe and the U.S. In contrast, a similar device, the speed bump, has had a slightly different evolution. A speed bump is also a raised pavement area across a roadway. They are typically found on private roadways and in parking lots. It is also noteworthy that there is a tendency not to have consistent design parameters from one installation to another. This type of device, regularly, has a height of 3–6 in (7.6–15 cm) with a travel length of 1–3 ft (0.3 m to 1 m). The speed of a car can be reduced to 8 km/h (5 mph). Nevertheless, as a result of not adhering to the law, in developing countries, these street artifacts tend to be much higher, and their dimensions are arbitrarily increased. The latter means that the height and travel length of the bumps are enlarged until cars actually slow down. This is partly justified by the fact that speed bumps have been routinely installed on private roadways and in parking lots without appropriate engineering investigations regarding their design and placement even in developed countries [32].

In Table 2, relevant concerns about bumps and humps are compared. The parameters in Table 2 were taken from USA [33–35] and Europe [36,37] references, but even locally, there are variations and updates. To deal with these discrepancies, the maximum and

minimum values were considered to provide a wider idea. The shapes and dimensions of the introduced bumps and humps are shown in Figure 1.

**Table 2.** Contrasts between bumps and humps.

| Issue | Bumps | Humps |
| --- | --- | --- |
| Length of profile | 0.3–1 m (1–3 ft) | 3.7–4.8 m (12–16 ft) |
| Width | 0.6–1.2 m | 4–8 m |
| Height | 3–15 cm | 2.5–15 cm |
| Profile | Circular/parabolic | Circular/parabolic/sinusoidal/Flat |
| Forced speed | 8 km/h (5 mph) | 12.5–40 km/h (15–25 mph) |
| Control action | Aggressive | Gradual |

After reviewing all this information, the authors arrived at the conclusion that, in spite of having federal or state laws, which have certainly progressed, in particular in developed countries, it has not been possible to completely regulate the actuation of GCs, LSVs, and PSTs in all environments. In addition, wanting to support the latter, the assistance of speed-control devices was briefly reviewed together with related laws, trying to guarantee a better vehicle–driver performance. However, we also took into account that traffic engineering studies may indicate that these calming devices would be unsafe at certain locations. Thus, a new proposal had to come, and this is an idea that has to be justified. Since the problem of all these vehicles is the driver's license regulation (which may imply a good/bad person's development) and given that certain constrained environments retain humps/bumps, it is worth automating the way these carts pass over these speed-control devices. As explained in the Introduction, the herein proposed system will automatically guide a car, and at the moment of arriving at a hump/bump, the car will slow down, then the car's frame will be elevated above the hump/bump level (to gently transport elderly or handicapped people), and the vehicle will continue its way to the end. From the desired behavior of the suspension system, the following constraints considered for this work are listed:

- Although the proposed design is applicable to any car suitable to be modeled as explained in Section 4, GCs or LSVs were considered as the to-be-controlled systems;
- The vehicle will be driven off road;
- The confined way will be equipped with speed bumps or speed humps as calming control devices;
- In order to automatically guide the vehicle, there will be a guide path along the car way. The path following will be performed via a sensor;
- World-wide control systems experience demonstrates that path following is not a challenge; that is why this study does not emphasize that;
- There will be an additional sensor to detect the speed hump/bump in order to warn about elevating the car chassis in time.

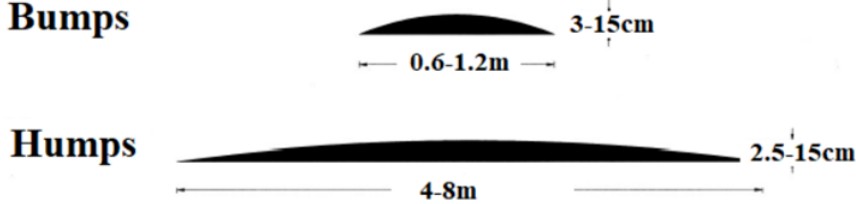

**Figure 1.** Without being exhaustive in the presentation calming control devices, the main differences between speed bumps/humps are sketched. See also Tables 1 and 2.

Thus, the analytical model and control law are explained next.

### 3.2.2. Potholes

Obviously, potholes also are an issue for the vehicle. If the car faces this problem, the control system will lift the chassis in order to avoid it. Offering a deep study about potholes is out of the scope of this manuscript, but this is a potential issue to consider, in particular in developing countries. Informally, a pothole is a pavement defect that can significantly affect a car's tires and structure. This road alteration was formally defined in [38] as a localized distress in an asphalt-surfaced pavement resulting from the breakup of the asphalt surface and possibly the asphalt base course. Pieces of asphalt pavement created by the action of climate and traffic on the weakened pavement are then removed under the action of traffic, leaving a pothole.

Most potholes are produced as a result of insufficient pavement thickness, poor drainage, failures at utility trenches and castings, and multifarious defects such as cracks left unsealed or unmaintained [38,39] (U.S. and Canada regulations). A very comprehensive manuscript is available on structural road policies in Europe in [40]. A report about Mexico City is available in [41]. See Figures 2 and 3.

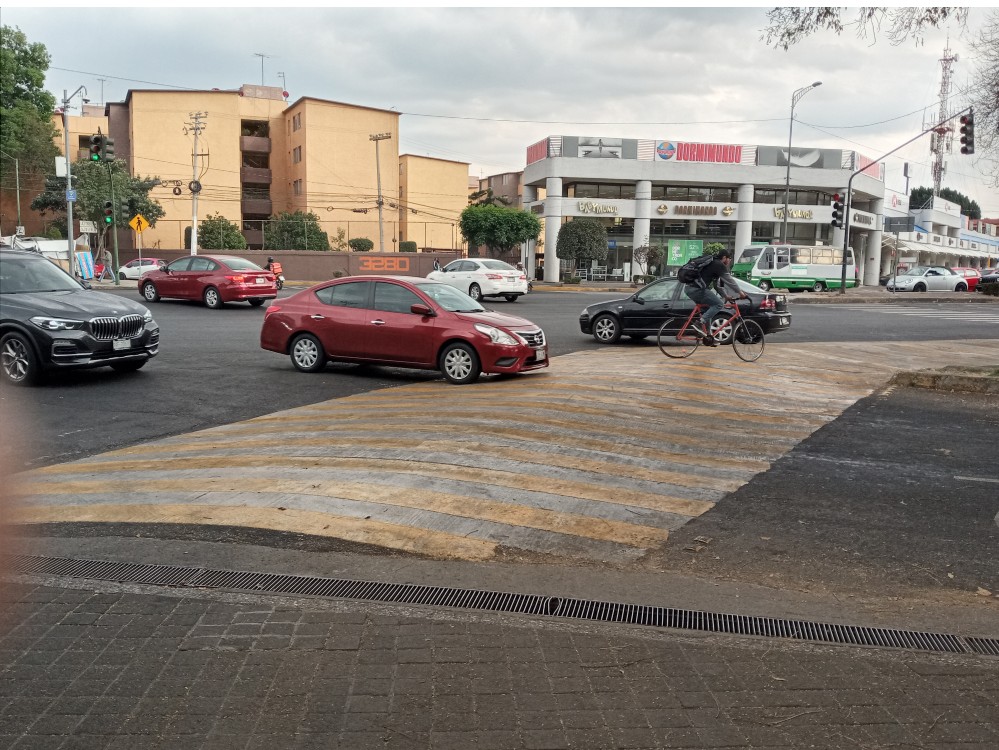

**Figure 2.** In some places of Mexico City, this huge speed hump is located, which had to be exaggerated in size to force drivers to slow down. Observe the car from a distance and the bicycle on the hump to have an idea of the dimensions of the obstacle. Although this image together with Figure 3 are on-road resources, the possibility of dealing with this in off-road environments for GCs and PSVs cannot be excluded.

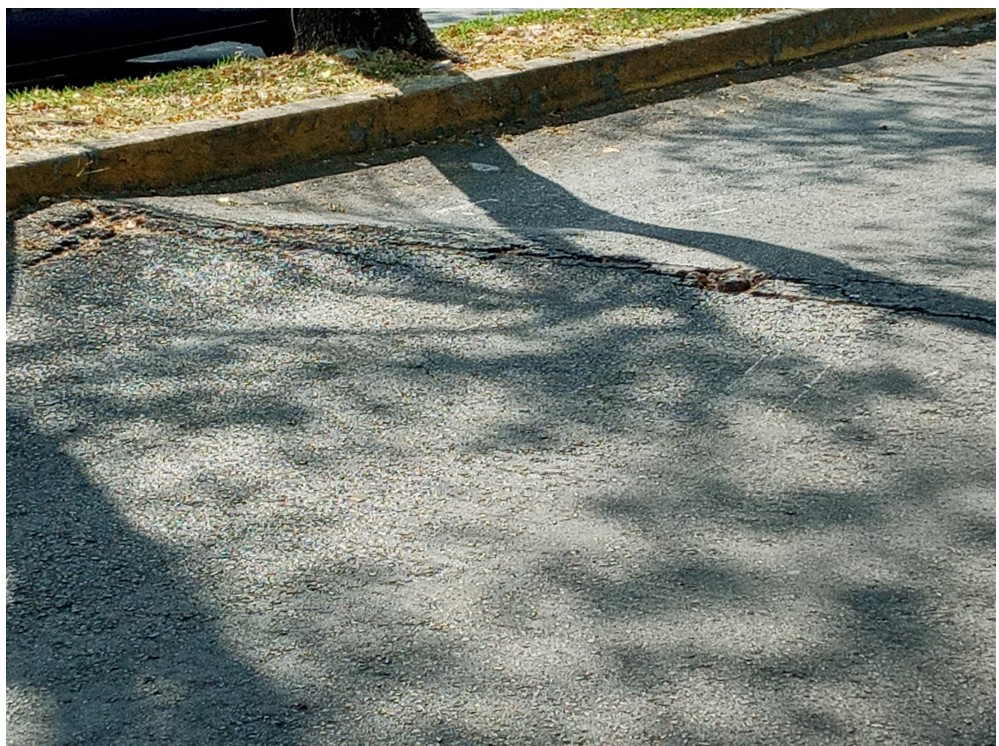

**Figure 3.** Somewhere in Mexico City. This is an example where a pothole turns into a "speed hump" caused by tree roots. Observe the sinking/uplift region.

## 4. Vehicular Suspension

A vehicular suspension consists of a set of elements between the suspended elements (chassis, engine, body, passengers, etc.) and the items that are not suspended (wheels, dampers, springs, brakes, and rigid bridges). In the context of this study, the functions of the suspension system are: to maintain, at all times, the vehicle's control by keeping the tires in contact with the road to ensure vehicle stability, to support the vehicle body, and to provide a comfortable ride for passengers by reducing vibrations caused by road irregularities.

A suspension is normally divided into a passive suspension (which consists of springs and dampers), a semi-active suspension (which uses a variable damper and common springs), and an active suspension (which employs hydraulics, air, or electric force actuators). Passive suspension is the simplest to design and economically advantageous. The main disadvantage of passive suspension is its limit of suppressing the vibration that occurs due to the uneven road surface. Semi-active suspension can vary the characteristics of the damper along with the road. Active suspension has the added benefit of negative damping and an increased range of force, which can be generated at low velocities. Nevertheless, this suspension needs a fully active actuator; therefore, significant energy input is required (the cost increases). For this reason and because of their higher reliability and affordable cost, semi-active suspensions are widely studied and applied to vehicle suspensions.

*Magnetorheological Damper*

A type of semi-active suspension works with magnetorheological dampers, which contain magnetorheological (MR) fluid. An MR fluid is typically 20–40% volume of particles of relatively pure elemental iron suspended within a liquid carrier such as mineral oil, synthetic oil, water, and/or glycol. In addition, commercial lubricants and additives to inhibit gravitational settling and reduce shock absorber wear are commonly added to magnetorheological dampers. It is important to mention that the fracture resistance of MR fluid depends on the square of the magnetic saturation of the suspended particles. Depending on the volume fraction of iron particles, the MR fluid may have a resistance

of a maximum yield strength between 30 kPa and 80 kPa to an applied magnetic field of 150–250 kA/m.

A magnetorheological (MR) damper usually contains: MR fluid, a sealing ring, two valve orifices, a coil, a diaphragm, and an accumulator. In the accumulator, there is nitrogen gas at a pressure of 20 bar, and the accumulator separates the MR fluid from the nitrogen gas. The valve port allows the MR fluid to pass up or down, and the sealing ring prevents friction. When current flows through the coil, it produces an electromagnetic field, and when the rheological fluid passes through the valves, the MR fluid changes its state to semisolid, so the damping force increases. Besides, MR fluid valves and the magnetic circuit are within the piston, and they regulate the flow of MR fluid within the damper. The current to the electromagnetic coil is injected through conductors found in the hollow shaft.

The control is performed by changing the force delivered to the suspension when the amount of current applied to the damper coil is modified. In other words, the current is the control input that modulates the MR damping fluid through the variation of a magnetic field, whereas the output is the force provided by the damper. As discussed below, these systems offer the reliability of passive devices, but also have the versatility and adaptability of active systems [42,43].

## 5. Literature Review

Currently, there are various devices that comprise the suspension system of a vehicle. Active and semi-active systems use many control techniques for two objectives, passenger comfort and vehicle stability. In the literature, optimal control, adaptive control, sliding mode control, fuzzy systems, generic algorithms, etc., can be found. Nevertheless, other control policies have been proposed specifically for suspension systems.

In [2], the control techniques Skyhook, Groundhook, and Mix-1-sensor for a semi-active suspension were compared. Such techniques are on–off, that is they only have two values corresponding to a soft or rigid damping force. Skyhook uses an imaginary damper between the body and the sky, trying to minimize the vertical acceleration of the vehicle body, but the contact between the tires and the ground is not very effective (stability control). Although this control strategy was proposed in [3] by D. Karnopp in 1974, it is a reference in control laws focused on comfort. Groundhook is similar to Skyhook, but focuses on the vehicle stability, reducing the dynamic forces between the tires and the surface using a virtual damper between the tire and the road. Furthermore, Mix-1-sensor minimizes the acceleration of the sprung mass by measuring the oscillation frequency of the suspension and thereby determining high or low damping.

On the other hand, in [44–47], the study and analysis of semi-active suspensions was reported. For instance, in [44], a system consisting of a variable damper manipulated by a PID controller, placed in series with a passive suspension, was studied. The controller function is to adjust the damping coefficient of the damper to keep the body always stable. As the output variable, the displacement of the suspension is used. For the model used in this article, a NARX model (based only on the use of sampled inputs and outputs) was assumed.

## 6. Mathematical Representations

### 6.1. Magnetorheological Damper as a Lumped, Hard Non-Linearity

Magnetorheological (MR) fluids are materials that respond to an applied magnetic field with a drastic change in the rheological structure. In the absence of a magnetic field, an MR fluid remains in the liquid phase. However, under a magnetic field, its viscosity varies according to the magnitude of the magnetic field applied up to the point of exhibiting solid-like characteristics. The strength of an MR fluid can be explained by the shear yield stress. The alteration of viscosity is continuous and reversible with respect to the applied magnetic field. As a consequence, MR fluid devices have the capacity of being fast-response interfaces between electronic devices and mechanical systems [42,43]. Although MR fluids are traditionally described by partial differential equations, from a lumped control systems

perspective, MR fluids can be investigated as hard non-linearities. MR fluid dampers are characterized by a large damping force and low power consumption. That is why they are used in vibration control systems.

Hence, to evaluate the potential of MR dampers in vibration control and observe the advantages, a model should be developed in order to represent their behavior. The main problem of MR dampers is their non-linear nature and the hysteresis of their force–velocity response. Therefore, an adequate characterization of an MR damper is a challenge. The model used in this work was a modified version of the one presented in [48] because it considers an accurate estimation of the damping force when the applied current is 0 A, as well as regarding: viscous damping, stiffness, and the non-linearities (friction and hysteresis). When the applied current is equal to zero, the force–velocity (FV) diagram is a sigmoidal curve accurately described by a hyperbolic tangent [48] (see Figure 4).

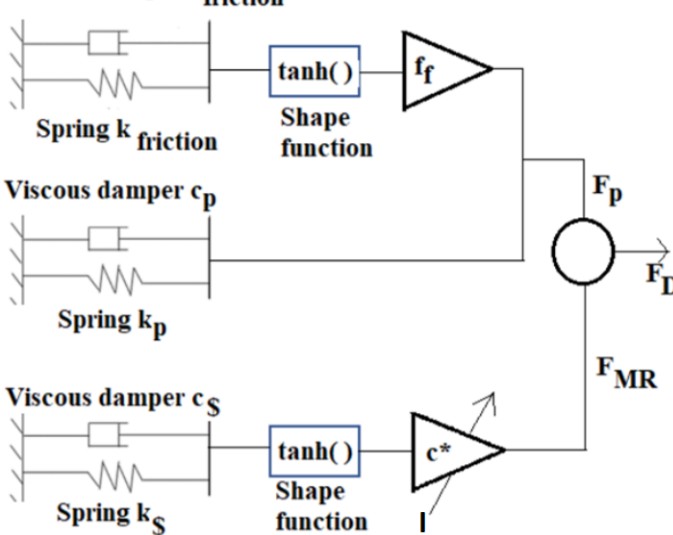

**Figure 4.** MR model. c* means $c_{MRpostyield}$.

The equations that model the MR damper in Figure 4 are:

$$F_D = F_P + F_{MR} \tag{1}$$

$$F_P = k_P z + c_p \dot{z} + f_f tanh(x_{friction} z + v_{friction} \dot{z}) \tag{2}$$

$$F_{MR} = c_{MRpostyield} I tanh(x_{preyield} z + v_{preyield} \dot{z}) \tag{3}$$

where:
$F_D$ is the MR damping force;
$F_P$ is the damping force when the current is 0 A;
$F_{MR}$ is the damping force due to the MR fluid changes;
$k_p$ is the stiffness coefficient;
$c_p$ is the damping coefficient;
$f_f$ is the friction coefficient;
$v_{friction}$ and $x_{friction}$ define the sigmoidal behavior of the friction component;
$c_\$ = c_{MRpreyield}, k_\$ = k_{MRpreyield}$;
$c_* = c_{MRpostyield}$ is the damping coefficient due to electric current;
$v_{preyield}$ and $x_{preyield}$ define the sigmoidal force component due to the MR fluid changes;
$z$ is the vertical damper displacement;
$I$ stands for the applied current in the MR damper.

### 6.2. Half-Car Suspension

In order to analyze the car suspension behavior, there are three models reported in the literature. Depending on what is required, a method to study and analyze the system is proposed, meaning a quarter-, half-, or complete-vehicle suspension.

Among these three representations, the quarter model is the most used for simulation because of its simplicity, but only vertical movement of the vehicle can be studied. The precision and accuracy of the model is increased with the half- and complete-vehicle models. The improvement comes from the fact that the rolling and pitching movements of the vehicle can be studied and the inertia moment of the vehicle can be included.

Therefore, in this paper, the half model of the suspension was considered, because the analysis focuses on studying the vertical behavior of the vehicle considering also the car speed. Another important point is that this model provides separate information from the front and rear suspension.

#### 6.2.1. Passive Car Suspension

The half-car model used in this study is shown in Figure 5, the MR damper's parameters are listed in Table 3, whereas the vehicle parameters are tabulated in Table 4.

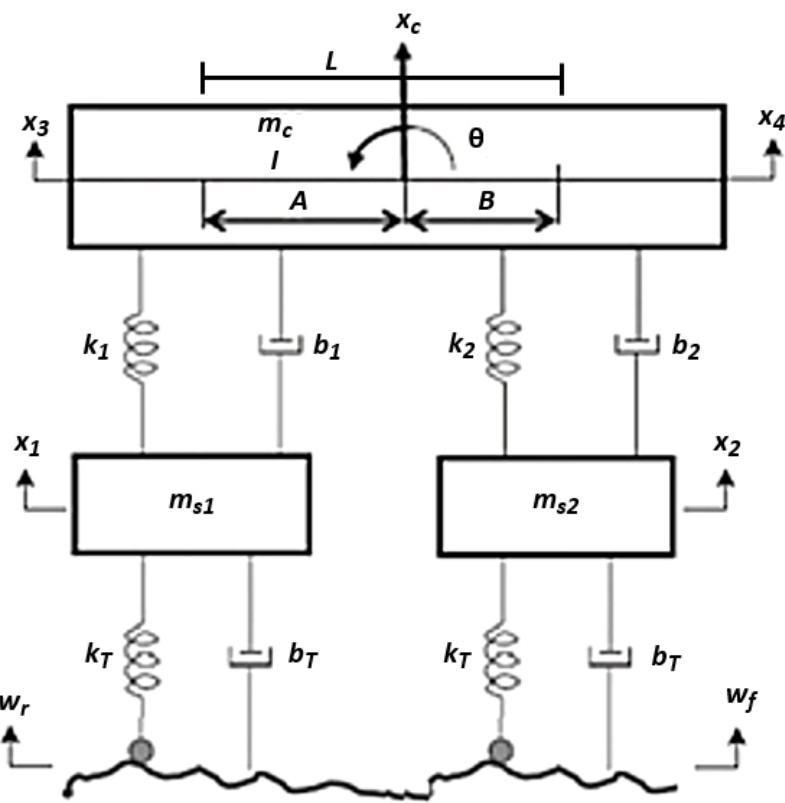

**Figure 5.** Half-car passive suspension scheme.

The equations that describe the model in Figure 5 are given by:

$$m_{s_1}\ddot{x}_1 = -k_T(x_1 - w_r) - b_T(\dot{x}_1 - \dot{w}_r) + k_1(x_3 - x_1) + b_1(\dot{x}_3 - \dot{x}_1) \tag{4}$$

$$m_{s_2}\ddot{x}_2 = -k_T(x_2 - w_f) - b_T(\dot{x}_2 - \dot{w}_f) + k_2(x_4 - x_2) + b_2(\dot{x}_4 - \dot{x}_2) \tag{5}$$

$$m_c\ddot{x}_3 = -k_1(x_3 - x_1) - b_1(\dot{x}_3 - \dot{x}_1) - k_2(x_4 - x_2) + b_2(\dot{x}_4 - \dot{x}_2) - Bm_c\ddot{\theta} \tag{6}$$

$$m_c\ddot{x}_4 = -k_1(x_3 - x_1) - b_1(\dot{x}_3 - \dot{x}_1) - k_2(x_4 - x_2) + b_2(\dot{x}_4 - \dot{x}_2) + Am_c\ddot{\theta} \tag{7}$$

$$I\ddot{\theta} = A[k_1(x_3 - x_1) + b_1(\dot{x}_3 - \dot{x}_1)] - B[k_2(x_4 - x_2) + b_2(\dot{x}_4 - \dot{x}_2)] \tag{8}$$

$$x_c = \frac{A x_3 + B x_4}{L} \tag{9}$$

$$\theta = \frac{x_4 - x_3}{L} \tag{10}$$

where:

$m_{s_1}$ = mass of the rear suspension (kg);

$m_{s_2}$ = mass of the front suspension (kg);

$m_c$ = mass of vehicle body (kg);

$k_1$ = Stiffness coefficient of the rear suspension (N/m);

$k_2$ = Stiffness coefficient of the front suspension (N/m);

$k_T$ = Stiffness coefficient of the front wheel (N/m);

$b_1$ = damping coefficient of the rear damper (Ns/m);

$b_2$ = damping coefficient of the front damper (Ns/m);

$b_T$ = damping coefficient of the tire (Ns/m);

$I$ = moment of inertia of the mass of vehicle body (kg m$^2$);

$A$ = distance from the center of mass to vehicle rear axle;

$B$ = distance from the center of mass to the vehicle front axle;

$L$ = distance between the front and rear axles;

$x_1$ = vertical displacement of the rear suspension (m);

$x_2$ = vertical displacement of the front suspension (m);

$x_3$ = vertical displacement of the rear of the vehicle (m);

$x_4$ = vertical displacement of the front of the vehicle (m);

$w_r$ = Rear wheel vertical displacement input (m);

$w_f$ = Front wheel vertical displacement input (m).

The employed magnetorheological damper's parameters are listed in Table 3.

**Table 3.** MR damper parameters.

| Parameter | $\dot{z} = \dot{x}_1 - \dot{x}_2 > 0$ | $\dot{z} = \dot{x}_1 - \dot{x}_2 < 0$ |
|---|---|---|
| $c_{MRpostyield}$ (N/A) | 873.9 | 828.4 |
| $v_{preyield}$ (s/m) | 7.7 | 9.6 |
| $x_{preyield}$ (1/m) | 12.2 | 15.3 |
| $c_p$ (Ns/m) | 530 | 925.2 |
| $k_p$ (N/m) | −7361.6 | −8941.1 |
| $f_f$ (N) | 115.3 | 31.7 |
| $v_{friction}$ (s/m) | 32.5 | 208.4 |
| $x_{friction}$ (1/m) | 66.7 | 609.9 |

The vehicle's parameters are listed in Table 4.

**Table 4.** Vehicle parameters.

| | |
|---|---|
| $m_c$ = 1794.4 (kg) | $k_{br}$ = 18,615 (N/m) |
| $I$ = 3443.05 $(N/A)$ | $k_{wf}$ = 101,115 (N/m) |
| $m_{wf}$ = 87.15 (kg) | $k_{wr}$ = 101,115 (N/m) |
| $m_{wr}$ = 140.14 (kg) | $L_f$ = 1.27 (m) |
| $c_{fb}$ = 1190 (Ns/m) | $L_r$ = 1.72 (m) |
| $c_{br}$ = 1000 (Ns/m) | $L$ = 2.99 (m) |
| $k_{bf}$ = 66,824.4 (N/m) | $F_{AD} = F_{AT}$ = 1500 (N) |

### 6.3. Proposed System

Taking into account Section 6.2.1, we now propose a semi-active suspension and a pneumatic actuator that work together with a Skyhook action. As explained in previous

sections, it is not an active suspension, since the actuator is not always in operation, but only when lifting is required due to some speed hump/bump. See Figure 6.

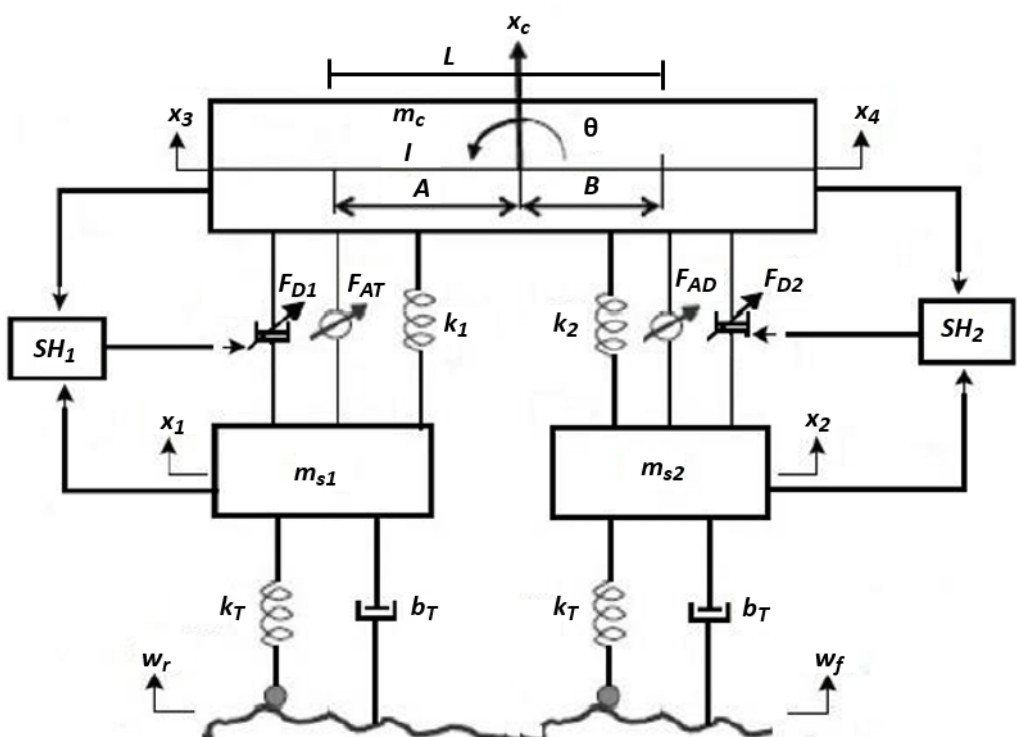

**Figure 6.** Half-car semi-active suspension model proposed to lift the vehicle chassis in the presence of humps/bumps.

The equations of motion derived from Figure 6 are the following:

$$
\begin{aligned}
m_{s_1}\ddot{x}_1 &= -k_T(x_1 - w_r) - b_T(\dot{x}_1 - \dot{w}_r) + k_1(x_3 - x_1) + b_1(\dot{x}_3 - \dot{x}_1) + F_{D1} - F_{AT} \\
m_{s_2}\ddot{x}_2 &= -k_T(x_2 - w_f) - b_T(\dot{x}_2 - \dot{w}_f) + k_2(x_4 - x_2) + F_{D2} - F_{AD} \\
m_c\ddot{x}_3 &= -k_1(x_3 - x_1) - k_2(x_4 - x_2) - F_{D1} - F_{D2} + F_{AT} - Bm_c\ddot{\theta} \\
m_c\ddot{x}_4 &= -k_1(x_3 - x_1) - k_2(x_4 - x_2) - F_{D1} - F_{D2} + F_{AD} + Am_c\ddot{\theta} \\
I\ddot{\theta} &= A[k_1(x_3 - x_1) + F_{D1} - F_{AT}] - B[k_2(x_4 - x_2) + F_{D2} - F_{AD}]
\end{aligned}
\tag{11}
$$

Above, $F_{D1}$ and $F_{D2}$ are defined as:

$$
\begin{aligned}
F_{D1} &= k_p(x_3 - x_1) + c_p(\dot{x}_3 - \dot{x}_1) \\
&+ f_f \tanh[x_{friction}(x_3 - x_1) + v_{friction}(\dot{x}_3 - \dot{x}_1)] \\
&+ Ic_{MRpostyield} \tanh[x_{preyield}(x_3 - x_1) + v_{preyield}(\dot{x}_3 - \dot{x}_1)]
\end{aligned}
\tag{12}
$$

$$
\begin{aligned}
F_{D2} &= k_p(x_4 - x_2) + c_p(\dot{x}_4 - \dot{x}_2) \\
&+ f_f \tanh[x_{friction}(x_4 - x_2) + v_{friction}(\dot{x}_4 - \dot{x}_2)] \\
&+ Ic_{MRpostyield} \tanh[x_{preyield}(x_4 - x_2) + v_{preyield}(\dot{x}_4 - \dot{x}_2)]
\end{aligned}
\tag{13}
$$

$F_{AD}$ = front actuator force, $F_{AT}$ = rear actuator force, and $x_c$, $\theta$ are defined as in Equations (9) and (10), respectively. Moreover, $SH_1$ represents the Skyhook controller for the rear suspension, whereas $SH_2$ is the Skyhook controller for the front suspension.

## 7. Control

Suspension control focuses on maintaining stability and passengers' safety, in particular if people are considered vulnerable. Linked to passengers' safety is comfort. It is well known that Skyhook control is a good approach when reducing vertical accelerations of the chassis is the main priority, but keeping passengers safe is also important. The damper is considered as being hooked to a fixed point in the sky. Notice that Skyhook control

is a switching law that commutes between two conditions, changing the structure of the controlled system at one's convenience (see Equation (14)).

The Skyhook control law is:

$$C_{SH} = \begin{cases} C_H, \dot{z}_s(\dot{z}_s - \dot{z}_{us}) \geq 0 \\ C_L, \dot{z}_s(\dot{z}_s - \dot{z}_{us}) < 0 \end{cases} \tag{14}$$

where $C_H$ stands for the high damping coefficient, $C_L$ represents the low damping coefficient, $z_s$ is the vertical displacement of the vehicle body (sprung mass), and $z_{us}$ stands for the vertical displacement of the suspension (unsprung mass). This law applies a low damping force when the chassis moves down, i.e., $z_s - z_{us} < 0$, and when the car frame displaces up, i.e, $z_s - z_{us} \geq 0$, the action is opposite. Therefore, this law has only two possible modes for the damping force. The strength of this force depends on the current supplied to the MR damper. It is important to mention that the saturation phenomenon exhibited in the actuator is a limitation in the control inputs. Besides, bumps/humps could be considered as disturbances to the system that could be decoupled according to [49,50].

*Stability Analysis*

Remember that a suspension model was proposed in Section 6.3 to achieve the control goal (informally established as lifting the car 12–20 cm above a velocity hump/bump). That system proposed contains actuators and MR dampers, and it is controlled by the switching Skyhook law. For this purpose, notice that the system described in Section 6.3 consists of a linear and a non-linear part. The latter contains the hyperbolic tangent of complicated terms. However, it is easy to see that that system can be written as:

$$\dot{x} = Ax + Bu + \Phi(x) \tag{15}$$

where $A \in \mathbb{R}^{m \times m}$, $B \in \mathbb{R}^{m \times n}$, $x \in \mathbb{R}^{m \times 1}$, $u \in \mathbb{R}^{n \times 1}$, and $\Phi(x) \in \mathbb{R}^{m \times 1}$.

$\Phi(x)$ represent the non-linear terms. The Lyapunov function proposed together with its time derivative along $x$ are:

$$V = x^T P x \tag{16}$$

$$\dot{V} = x^T(PA + A^T P)x + 2x^T P\Phi(x) + 2x^T Bu \tag{17}$$

$$-Q \triangleq PA + A^T P \tag{18}$$

$$\dot{V} = -x^T Q x + 2x^T P(\Phi(x) + Bu) \tag{19}$$

To find conditions for the negative definiteness of $\dot{V}$, $Bu$ is defined as follows:

$$Bu = -\Phi(x) + \eta \tag{20}$$

$$\dot{V} = -x^T Q x + 2x^T P \eta \tag{21}$$

$$\eta \triangleq RxK \tag{22}$$

$$\dot{V} = -x^T Q x - (2x^T PRx)K \tag{23}$$

Above, $R \in \mathbb{R}^{m \times m}$ and $K \in \mathbb{R}$. After including the controller, the algebra is developed, and it was observed that the term $(2x^T PRx)$ has to be equal to $x_2^2 - x_2 x_4$. The later implies that:

$$\dot{V} = -x^T Q x - (x_2^2 - x_2 x_4)K \tag{24}$$

The latter is constrained to the fact that $x_2 x_4 > 0$. This result can be improved by a signum function considering $K$ as:

$$K \triangleq \frac{C}{|(x_2^2 - x_2 x_4)|}, x_2 \neq x_4 \tag{25}$$

$$C = \begin{cases} C_H \\ -C_L \end{cases} \quad C_H, C_L > 0 \tag{26}$$

where $||$ denotes the absolute value. Finally:

$$q \triangleq (x_2^2 - x_2 x_4) \tag{27}$$
$$\dot{V} = -x^T Q x - C sgn(q) \tag{28}$$

where the combined action of the signum function and $C$ allows $\dot{V} \leq 0$.

With the closed-loop suspension schematic and the control strategy explained, the block diagram of the complete control system is presented in Figure 7.

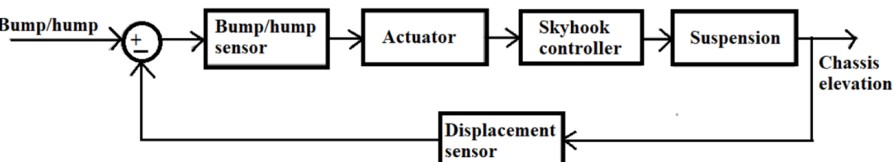

**Figure 7.** Real closed-loop system.

## 8. Simulation

To perform the simulations, the MATLAB/Simulink software was used. To approximate and simulate speed, the v = d/t formula was used. Then, if the vehicle speed were 5 km/h (1.38 m/s) and the wheelbase were 2.99 m, the rear suspension would pass the bump approximately 2.2 s after the front suspension does. That is why the control action is on two seconds before the car arrives at the hump. The moment at which the vehicle meets the hump is at second 6. The latter is registered by a proximity sensor (see Section 9). Although the following experiments were developed to contrast four ways to lift the chassis of the car, for the sake of exposition, the first case (passive suspension action) and the last one (the proposed design) are illustrated:

- Passive suspension. In Figure 8, oscillations are observed when the car reaches the obstacle and remains this way until the chassis goes down until reaching zero (normal body level). It is to be expected that the car has this behavior since the springs are oscillating and dissipating energy;
- Passive suspension plus actuators. In this case, many ripples showed up, negatively impacting the stable behavior of the structure;
- MR damper suspension. This case did not show many oscillations, but (as in the other) the suspension system just waited to reach the obstacle exactly at Second 6; the structure was elevated (then ripples appeared) and finally came down;
- MR damper+actuators. This case permits providing an in-advance action to lift the chassis at Second 2, but at this moment, a high overshoot of approximately 4 cm harmed the performance;
- Proposed system. MR dampers + actuators + Skyhook control action. This event is plotted in Figure 9. First, the speed bump is illustrated there to show the moment at which the car arrives at it. The proposed performance is provided also there, and it can be noticed because it has a big overshoot at Second 6 and many undesirable oscillations when the chassis leaves the obstacle. Although this performance was demonstrated to be better than the others cited above, it was afterwards improved by adding a smoother linear dynamics plus a PID controller. This fact definitely ameliorated the latter behavior, as can be observed there.

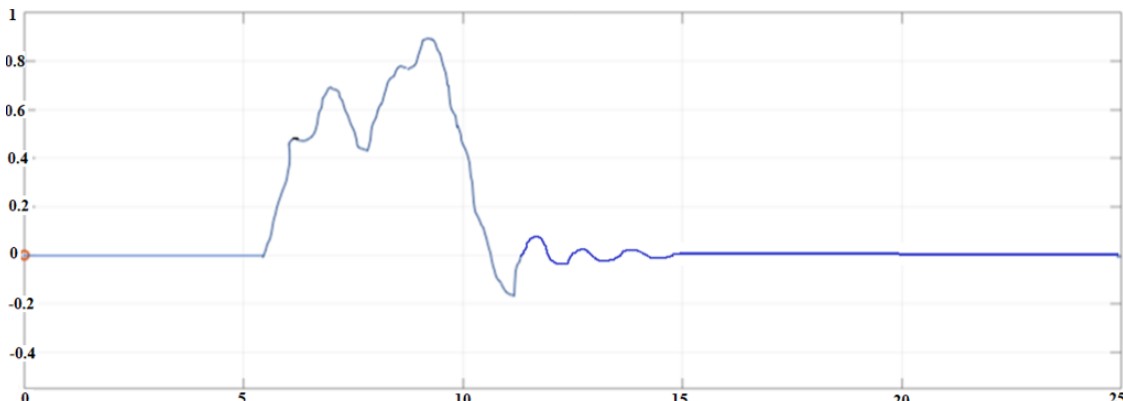

**Figure 8.** Passive actuation to elevate the structure of the vehicle when it meets a speed bump/hump. Although the goal was accomplished, there were undesirable ripples, and the motion of the chassis was abrupt.

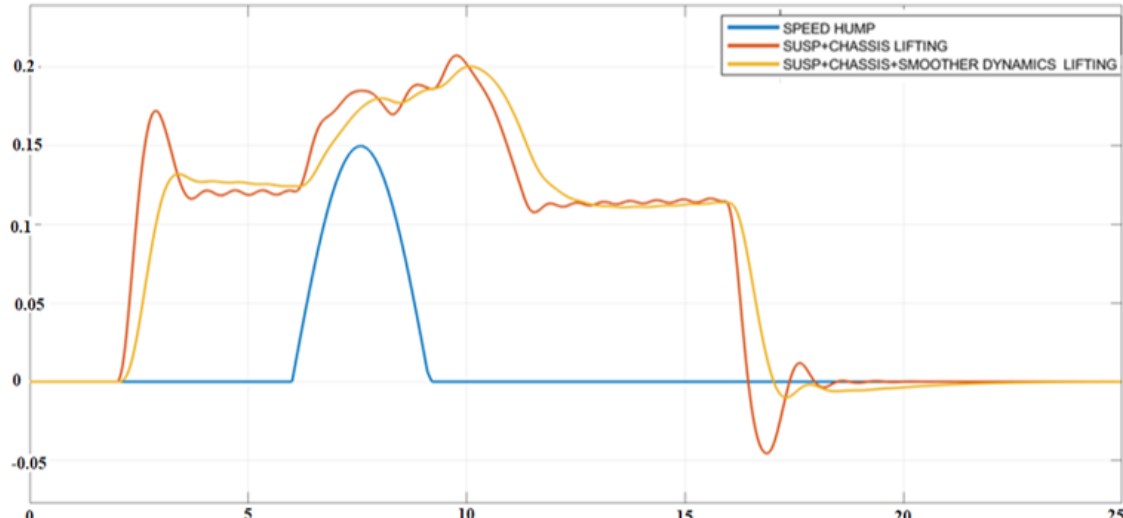

**Figure 9.** The proposed solution (red) and an ulterior version of it (yellow) could accomplish the task of lifting the car in advance (at Second 2, i.e., 4 s before reaching the bump). Besides, from Seconds 9–16, the car remains up for safety, and the return to its original height transits softly during all the vertical displacement. Notice that the return to its original position (zero level) was performed gently, and the hump was sinusoidal.

The case shown in Figure 9 uses a sinusoidal model for the speed-calming device. According to the figure, it is a speed bump. The system reacts differently if the vehicle meets a Gaussian bump. As shown in Figure 1 and Table 2, the way a bump and a hump force cars to slow down changes. Humps can be higher than humps (remember Figure 2). If the height of a Gaussian hump is increased, the controller performs well, as illustrated in Figure 10. Observe that the chassis was elevated beforehand, and once on top of the speed hump, the control produced an even higher vertical displacement to avoid the bump crest. The return to the original level of the chassis was also smooth.

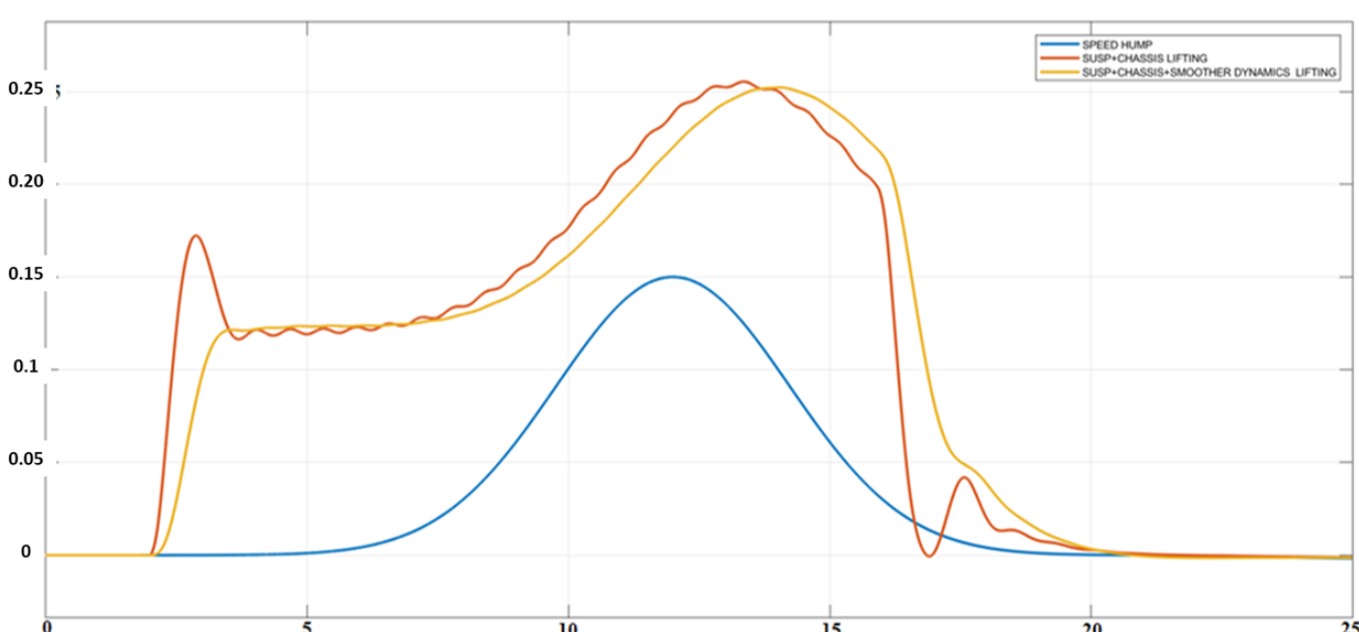

**Figure 10.** Positive reaction of the herein designed system against a Gaussian speed hump. In this image, the hump is illustrated. It starts at the 5th second and finishes at the 20th second, and the elevation is 15 cm. Notice that the original proposal exhibited an overshoot, but the ulterior improvement canceled this, as well as other undesirable oscillations.

An additional example is offered, modifying the smoother dynamics used for the case shown in Figure 9. It was also possible to elevate the car when encountering a pothole, and this situation is illustrated in Figure 11. As indicated before, the control action actuated 2 s in advance. The pothole was modeled by a Gaussian function that represented a hole that had a minimum depth of 10 cm = 0.1 m. The hole encompassed the time from 5 s to 20 s. The reaction of the car was right, slowly descending with a small overshoot.

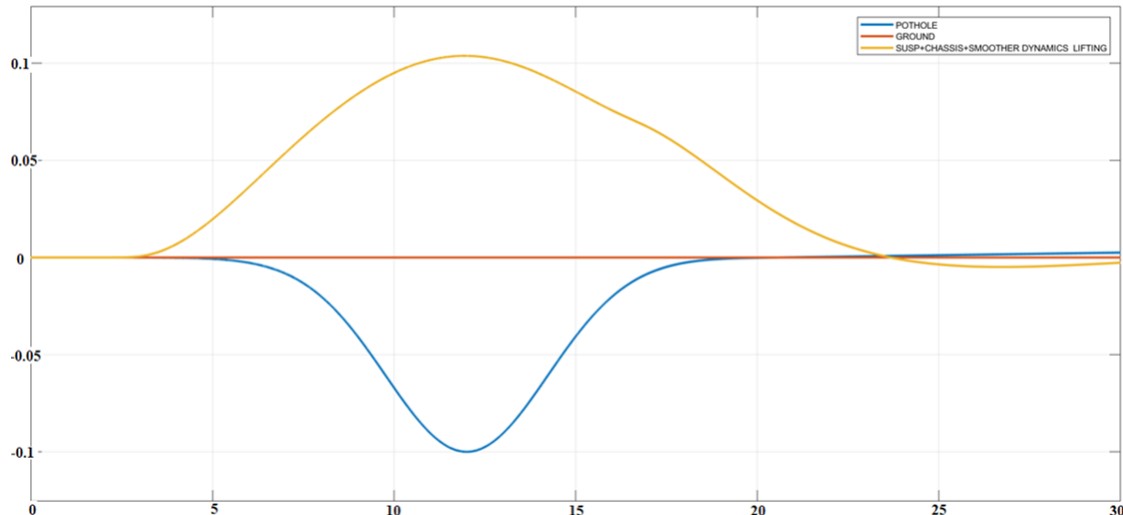

**Figure 11.** Controlled action to elevate the structure of the vehicle before falling in a pothole that was modeled by a Gaussian function.

Particular measures could be acquired with the help of sensors. For that, physical considerations are offered next in Section 9. In the case of having another car, the parameters should be modified, but also, a more robust controller is possible to be designed.

## 9. Physical Considerations

The successful simulations considered a car vehicle, and this methodology can be adapted to almost any vehicle subject to the constraints given in Section 3.2. The proposed suspension consisted of MR dampers, passive elements, and the control system. The minimum value of the current was 0.2 A, while the maximum value was 0.4 A. The force provided by the actuators was 2300 N, with which the bodywork moved approximately 12–20 cm.

### 9.1. Commercial Systems to Lift the Chassis of a Car

As for commercial vehicle-lifting devices today, there are systems that can lift the vehicle by pneumatic actuators with one unit control panel placed on the vehicle dash [51], the disadvantage of this being that the the driver has to manually adjust it to the height he/she wants; of course, there are limits to the change in height. Another example are the so-called *coilovers* [52]. These devices consist of a damper, with a coil spring encircling it. The damper and spring are assembled as a unit prior to installation and replaced as a unit when either of the sub-components fails. These mechanisms can modify the height of the vehicle by varying the length of its springs using a nut with hand tools. In addition, the wheels must be removed to access the thread. A modification of this is a coilover system with an in-dash control unit for the vehicle; the height is regulated by four motors to turn the thread and, with this, vary the height.

In addition, the complete proposal is that the car can avoid a speed-calming device sudden hit, but also, in order to eliminate the need for a human driver, a path following for this car was considered. Without neglecting the importance of this part, the literature revision is focused on the novelty. In Figure 12, the idea of the automated path following is provided. In [53], a detailed description of this concept was developed, proposing a microcontroller unit and sensors. It was assumed in advance to have sensors for the path following. In addition, sensors to measure the distance from the car to the bumps/humps, as well as sensors to detect speed-calming devices' heights are briefly described in the next section. See also [54].

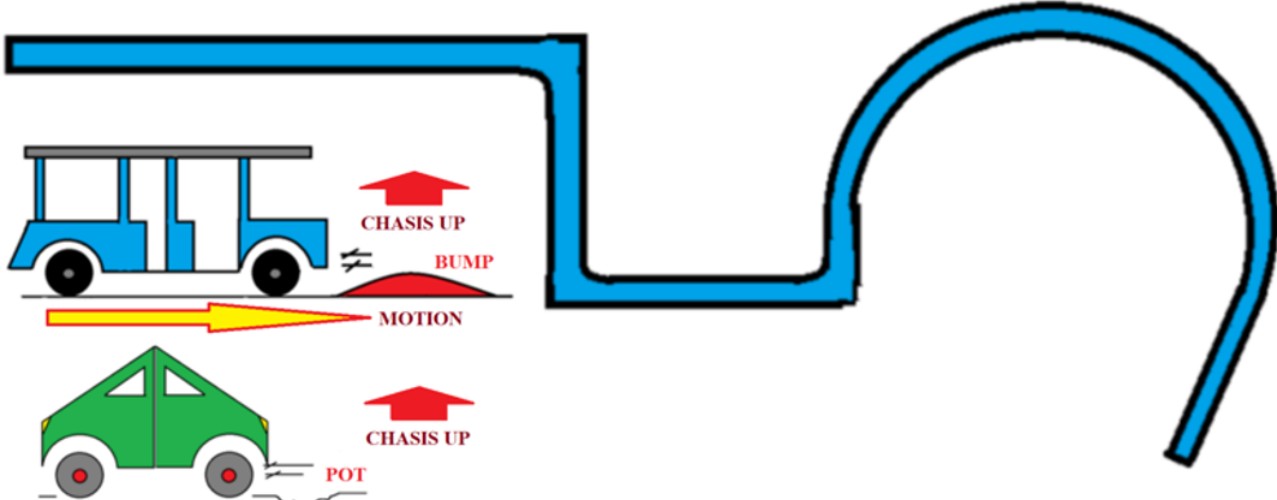

**Figure 12.** Path following idea to automate the vehicle trajectory.

### 9.2. Commercial Sensors

There are two types of probes to use here: distance and height detectors. The former allow us to measure the car–obstacle distance to determine when to slow down. Two seconds before that encounter, the Skyhook controller works to elevate the chassis. The height of the bump is measured by a height device. These probes are photoelectric sensors based on light-based detection. Within this class, color sensors are key. This type of device

is useful when bumps/humps are painted yellow or white. The latter depends on the regulations of each country/region. When the distance to the target changes, the received light intensity also changes, as happens when passing over speed-calming structures [55]. The latter source also described laser 2D displacement sensors. Their performance is enhanced by an easy setup plus data storage via a PC. For instance, the mid-range LJ-G080 sensor measures in the z-axis, $8 \pm 2.3$ cm, while the long-range version, LJ-G200, reaches $20 \pm 4.8$ cm [56]. Another type of apparatus was described in [57]. This family of components works according to a pulsed red laser diode. They are analog and RS 485.

Although the instruments just described may be adapted for vehicle applications, in [58], specialized optical laser height sensors were developed for car mounting. HF laser height sensors are designed for non-contact distance measurement for vehicle dynamics testing. These instruments range from 10–90 cm, depending on the type. They are produced according to the principle of optical triangulation. This means that a visible red laser is focused on the road surface. Then, a light is projected on the target, and the reflected light is collimated on a linear CCD matrix (a CCD device is an integrated circuit that contains a certain amount of bonded or coupled capacitors). The distance to the object is determined knowing the position of the point of light on the CCD matrix. The sensor output is directly proportional to the measured height. The way to place sensors on a sedan car was explained in [58]. This rule has been proposed for GCs and LSVs (see Figure 13).

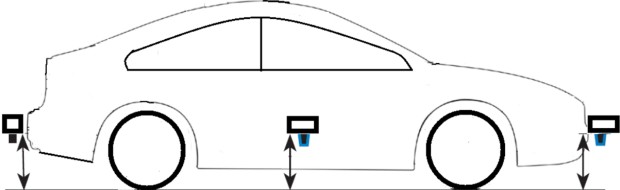

**Figure 13.** Distribution of sensors in a sedan car according to [58]. In this proposal, the plan is to follow that rule for the front detector in the automated vehicles.

*9.3. Other Vehicles*

Working with comparable passenger vehicles is also possible as claimed in Figure 14. For instance, the MotoEV Electro Transit Buddy 9 Passenger Shuttle [59] is characterized by the following parameters:

- Model: MotoEV Electro Transit Buddy 9 Passenger Shuttle;
- Passenger capacity: 9 people;
- Vehicle dimensions: $13.9'$ L $\times$ $4.88'$ W $\times$ $6.72'$ H;
- Gross vehicle weight: 3855 Lbs ($\approx$1750 kg);
- Speed: up to 25 mph;
- Range: up to 50 miles (full capacity);
- Climb: 20% grade (full capacity);
- Electric motor: 5 KW (6.7 Hp);
- Maxload: 2100 Lbs ($\approx$953 kg);
- Ground clearance: $7''$ ($\approx$17.7 cm);
- Batteries: QTY 12-Trojan 105 or US2200;
- Suspension: independent/rear steel plate suspension;
- Drive: rear-wheel drive.

Without details, the set of these parameters are approximate to the ones shown in Table 4, and adjustments can be made in the designs to produce results as those already offered here.

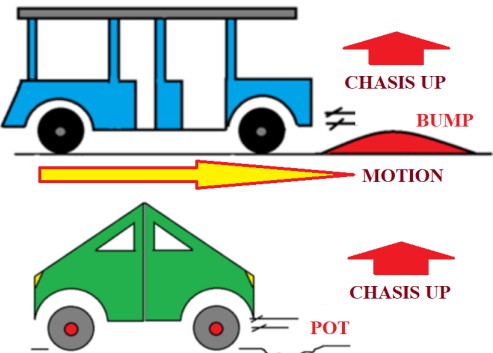

**Figure 14.** The frame of the car will be lifted up by a control action to deal with speed humps and speed bumps.

## 10. Preparing for the Future of Automated Transportation: Off-Road Displacement of Vulnerable People

At the beginning of this document, the authors tried to motivate the real need for autonomous vehicles, type GC or PSV, that can transport vulnerable passengers according to the constraints given in Section 3.2. It was shown that not only legislation about driver's license holding for GCs and PSVs is an issue, but also the fabrication standards of such vehicles. Furthermore, the principles and codes for placing off-road speed-calming devices plus the random emergence of pot holes induce the action of automating the car. In spite of the fact that this work was devoted to automatically dealing with bumps/humps by the elevation of the framework of the car, it was remarked that the path following will be solved elsewhere because this problem is well known. In this short section, the intention is to emphasize the future (not very far away) implications of automated cars. Besides, it is also remarked that (at least) the U.S. Government is already taking actions on the matter [60]. Among the many implications of automation, in [60], it was reflected that automation represents tremendous possibilities for improving the mobility of travelers with disabilities, as remarked in the present study.

Through the Accessible Transportation Technologies Research Initiative (ATTRI), the Department of Transportation fosters initiatives with various government organizations in conjunction with the disabled community to develop research that allows this vulnerable group to travel comfortably and safely. It is also important to mention that disability involves not only mobility, vision, and hearing, but also cognitive impairment. Special needs is defined as an individual with a mental, emotional, or physical disability.

Although the Society of Automotive Engineers (SAE; https://www.sae.org/ (accessed on 13 February 2022)) [60] identifies six levels of driving automation, the automatic suspension involves such a self-performed process, and the given definition about these terms are the following:

- Automation: Use of electronic or mechanical devices to operate one or more functions of a vehicle without direct human input. Generally applies to all modes;
- Automated vehicle: Any vehicle equipped with driving automation technologies (as defined in SAE J3016). This term can refer to a vehicle fitted with any form of driving automation (SAE Level 1–5);
- Automated Driving System (ADS): The hardware and software that are collectively capable of performing the entire dynamic driving task on a sustained basis, regardless of whether it is limited to a specific operational design domain. This term is used specifically to describe a Level 3, 4, or 5 driving automation system (SAE J3016).

The authors remark that this research effort is innovative in favor of disable passengers, in line with these precepts.

## 11. Conclusions

After reviewing the definitions and regulations of GCs, LSVs, and PSTs, as well as speed-calming devices in developed and developing countries, a complete review of the laws is still missing, and sometimes, there are conflicts between federal and state laws (in particular in the U.S.). Although Europe is a more complex territory in creating regulations, it is generally considered to have more uniform and congruent statutes about GCs, LSVs, and PSTs than other regions of the world. The case for developing countries is much more complicated, as expected. Remember that an important issue is allowing driving this type of transport without holding a valid driver's license. The suggested way to solve the complex problem above is by lifting the car's frame when the vehicle meets the obstacle (even a pot hoke). In reviewing all these laws, the purpose was to support the need for the proposed design. The next step in the investigation is to work on the implementation.

**Author Contributions:** Conceptualization, M.S.-O., R.Z.-Y. and R.A.R.-M.; Methodology, M.S.-O., R.A.R.-M. and L.C.F.-H.; software, R.Z.-Y.; validation, R.Z.-Y., M.S.-O., R.A.R.-M. and L.C.F.-H.; formal analysis, R.Z.-Y.; investigation, M.S.-O. and R.Z.-Y.; resources, R.A.R.-M. and L.C.F.-H.; data curation, R.Z.Y. and M.S.-O.; writing—original draft preparation, R.Z.-Y.; writing—review and editing, R.Z.-Y., M.S.-O. and L.C.F.-H.; supervision, R.Z.-Y. and R.A.R.-M. All authors have read and agreed to the published version of the manuscript.

**Funding:** The APC was funded by Tecnologico de Monterrey, School of Engineering and Sciences, through the GIEE-Robótica.

**Acknowledgments:** The authors are grateful to Tecnologico de Monterrey, School of Engineering and Sciences for having supported this research and for having financed the cost of the publication.

**Conflicts of Interest:** The authors declare no conflict of interest.

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
