# Peer review of "Soft Passing over Traffic-Calming Devices by Controlled Suspension in Low-Speed Robotic Vehicles for Vulnerable People"

_applsci, doi:10.3390/app12063109_

Round 1

Reviewer 1 Report

Remarks are provided in a separate pdf file.

Author Response

Thanks to all reviewers for their valuable comments.

Reviewer 2 Report

The contribution of this work is very good and deserves publication. The title has been formulated unambiguously conveying the focus of the study.

Appropriate research goals are chosen in this contribution, which shows that the authors have a high level of understanding of current research within the field of their research. The authors successfully used the appropriate techniques for analysis of the research objects.

The accurate interpretation  of outcomes, well substantiated by the results of the analysis has been achieved by them. The presentation of the results in terms of the research objectives has been successfully made.

The authors have been able to draw logical conclusions from the results. The results and their respective discussion points prove the efficacy of the proposed method. Conclusions are accurate and clearly based on outcomes.

Try to address the problem of the constraints (limitations) of the inputs.

In fact, one of the most important problem is the limitation of the inuts which does not allows to realize a controller properly.

In the following litterature you can fiund some possible Inspiration to discuss and to address this Problem at least in the literaturee.

Prattichizzo, P. Mercorelli On some geometric control properties of active suspensions systems, January 2000, Kybernetika -Praha-36(5)

D. Prattichizzo, P. Mercorelli, A. Bicchi and A. Vicino, "Geometric disturbance decoupling control of vehicles with active suspensions," Proceedings of the 1998 IEEE International Conference on Control Applications (Cat. No.98CH36104), 1998, pp. 253-257 vol.1, doi: 10.1109/CCA.1998.728386.

Y. Su et al."Global Finite-Time Stabilization of Planar Linear Systems With Actuator Saturation," in IEEE Transactions on Circuits and Systems II: Express Briefs, vol. 64, no. 8, pp. 947-951, Aug. 2017, doi: 10.1109/TCSII.2016.2626199.

Author Response

Thank you for your kind assistance.

Reviewer 3 Report

The authors have come up with a novel idea which is highly needed for the welfare of people in the society. The paper is very well written. The simulation results based on the design they have considered clearly provides a comfortable means of transportation for the patients to be transported comfortably in the vehicles which is of utmost importance. 

Please check the clarity of Figure. 6.

The graphical results from Figure.8 and Figure. 9 clearly manifests the efficacy of the proposed idea. This could also be applied in several developing countries where need of such an idea is high essential. 

I recommend the paper towards publication with minor revisions of changing the Figure. 6 alone. Good job authors.

Author Response

Thank you for your valuable assistance.

Round 2

Reviewer 1 Report

I still do not agree with some of your comments, in particular those related to sections of the text that have only minor relevance to the subject. However, final responsibility for the content lies with the authors and reviewer's notes might only help to improve the writing.